# A Review on Applications of Artificial Intelligence in Wastewater Treatment

Yi Wang [1,2,3,4], Yuhan Cheng [3,5], He Liu [6], Qing Guo [3,7], Chuanjun Dai [3,4], Min Zhao [3,4,*] and Dezhao Liu [1,2,*]

1 Institute of Agri-Biological Environmental Engineering, College of Biosystems Engineering and Food Science, Zhejiang University, Hangzhou 310058, China; yiwang@wzu.edu.cn
2 Key Laboratory of Equipment and Informatization in Environment Controlled Agriculture, Ministry of Agriculture and Rural Affairs, Key Laboratory of Intelligent Equipment and Robotics for Agriculture of Zhejiang Province, Hangzhou 310058, China
3 School of Life and Environmental Science, Wenzhou University, Wenzhou 325035, China; chengyh294@163.com (Y.C.); guoqingwzu@163.com (Q.G.); chuanjdai@wzu.edu.cn (C.D.)
4 National & Local Joint Engineering Research Center for Ecological Treatment Technology of Urban Water Pollution, Wenzhou University, Wenzhou 325035, China
5 Key Laboratory of Coastal Environment and Resources of Zhejiang Province, School of Engineering, Westlake University, Hangzhou 310030, China
6 School of Mathematics and Physics, Wenzhou University, Wenzhou 325035, China; liuhe900418@163.com
7 Environmental Engineering Program, University of Northern British Columbia, Prince George, BC V2N 4Z9, Canada
* Correspondence: zmcn@tom.com (M.Z.); dezhaoliu@zju.edu.cn (D.L.)

**Abstract:** In recent years, artificial intelligence (AI), as a rapidly developing and powerful tool to solve practical problems, has attracted much attention and has been widely used in various areas. Owing to their strong learning and accurate prediction abilities, all sorts of AI models have also been applied in wastewater treatment (WWT) to optimize the process, predict the efficiency and evaluate the performance, so as to explore more cost-effective solutions to WWT. In this review, we summarize and analyze various AI models and their applications in WWT. Specifically, we briefly introduce the commonly used AI models and their purposes, advantages and disadvantages, and comprehensively review the inputs, outputs, objectives and major findings of particular AI applications in water quality monitoring, laboratory-scale research and process design. Although AI models have gained great success in WWT-related fields, there are some challenges and limitations that hinder the widespread applications of AI models in real WWT, such as low interpretability, poor model reproducibility and big data demand, as well as a lack of physical significance, mechanism explanation, academic transparency and fair comparison. To overcome these hurdles and successfully apply AI models in WWT, we make recommendations and discuss the future directions of AI applications.

**Keywords:** artificial intelligence; wastewater treatment; machine learning; artificial neural network; search algorithm; water quality





## 1. Introduction

Water resources, one of the most significant elements in human life and production processes, are now under serious threat from harmful pollutants caused by human activities and natural processes [1]. A large amount of wastewater is produced every day, most of it contains toxic pollutants and is directly released into the environment without being treated or reused [2]. In general, untreated sewage is rich in various nutrients, organic matters, suspended solids (SSs), organic micropollutants and pathogenic and nonpathogenic microorganisms [3]. Depending on the source, wastewaters can be classified into six categories: municipal, domestic, industrial, medical, agricultural and nuclear. Among them, municipal and domestic wastewaters are the most abundant, and the research on WWT is concentrated on these two types of sewage [4]. Because wastewaters with different sources

may differ significantly and have different physical and chemical properties, it is important to assess their characteristics before choosing the appropriate treatment process. In order to protect the limited water resources, environment and human health and meet the growing water demand, it is imperative to explore the treatment of wastewaters and their reuse as a resource. Wastewater treatment (WWT) removes contaminants from sewage involving a combination of physical, chemical and biological processes [5], and produces clean water that is safely released back into the environment. The key issues of WWT are how to reduce water pollution to a safe level efficiently while producing fewer negative impacts on the environment and decreasing energy consumption. Overall, WWT, as an indispensable process for water resources reuse and sustainable development, is essential for protecting public health and the environment and has been widely used in industrial and agricultural fields [6]. Effective and innovative technologies are urgently needed to improve efficiency, reduce cost and decrease the energy consumption of WWT [7,8].

Artificial intelligence (AI) refers to the ability of a computer program to realize autonomous learning, reasoning, judgment and decision making by simulating human intelligence. AI, one of the most impressive inventions during this century, is developing rapidly and has been widely applied in many areas such as natural language processing (NLP), computer vision (CV) and autopilot. Benefiting from its high efficiency, AI can be used for classification and regression analysis of massive amounts of data generated anytime and anywhere, thus energizing industries and promoting the development of all walks of life greatly. With the rapid development of computer technology, machine learning (ML), as an important branch of AI, uses data, algorithms, statistics and mathematical optimization to imitate the way that humans learn, gradually improving its accuracy and achieving artificial intelligence. Due to the advent of the big data era and increasingly strong supercomputing capabilities, ML is becoming more and more popular and has been successfully implemented in industry, agriculture, medicine, environmental protection, scientific research and other fields.

The WWT process mainly consists of water quality monitoring, laboratory-scale research and process design. AI models are becoming more and more popular in wastewater-related fields, especially in recent years (see Figure 1), and have been employed for the prediction and optimization of the WWT process [9,10]. In previous WWT-related research, AI models have shown very good prediction and optimization performances [11], and have been successfully applied to WWT process design [10,12], water quality monitoring [13,14], WWT process parameters optimization [15,16] and WWT process performance prediction [17,18]. These pieces of research have demonstrated that an AI model, as a powerful tool, has achieved great success in the applications of WWT-related fields. However, most of the review works of AI applications in WWT focus on some technique or process design aspects of WWT, such as adsorption processes, membrane bioreactors, membrane processes and WWTP. A comprehensive review of AI applications in the WWT process involving water quality monitoring, laboratory studies and real process design has rarely been seen until now. Additionally, there are few review articles available, which comprehensively introduce the commonly used AI models in WWT and summarize their advantages, disadvantages and proposals.

In this review, an overview of the literature on the applications of AI models and smart technologies, with a special focus on most ML in WWT, is presented. This review is not intended to cover all the applications of AI, ML and smart technologies in WWT, but rather to summarize the key findings of these important published works and analyze future development trends of AI in WWT. The WWT-related applications are mainly concentrated in the modeling, prediction and optimization of water and wastewater treatment processes, containing water quality monitoring, laboratory-scale research and process design. The remainder of this review is organized as follows: Section 2 systematically introduces the commonly used AI models in WWT and summarizes the strengths and weaknesses of each model. Section 3 outlines the applications of AI models in WWT, including water quality monitoring for data acquisition, laboratory-scale research and process design. Section 4

presents challenges and future perspectives of AI applications in WWT. Finally, Section 5 ends with a conclusion.

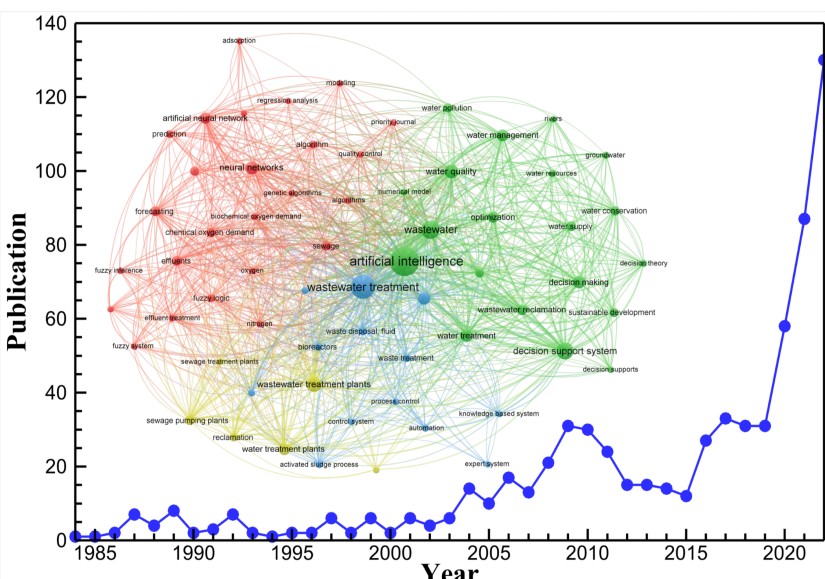

**Figure 1.** Number of publications related to artificial intelligence in wastewater. Number of publications was obtained by using Elsevier's Scopus database with queries TITLE-ABS-KEY (terms). Inset: network visualization of research topics related to artificial intelligence in wastewater, generated by VOSviewer based on keywords.

## 2. AI Models

The most commonly used AI models for WWT in the literature are shown in Figure 2. These models used for prediction and optimization can be classified into three main categories: Artificial Neural Network (ANN), Machine Learning (ML) and Search Algorithm (SA).

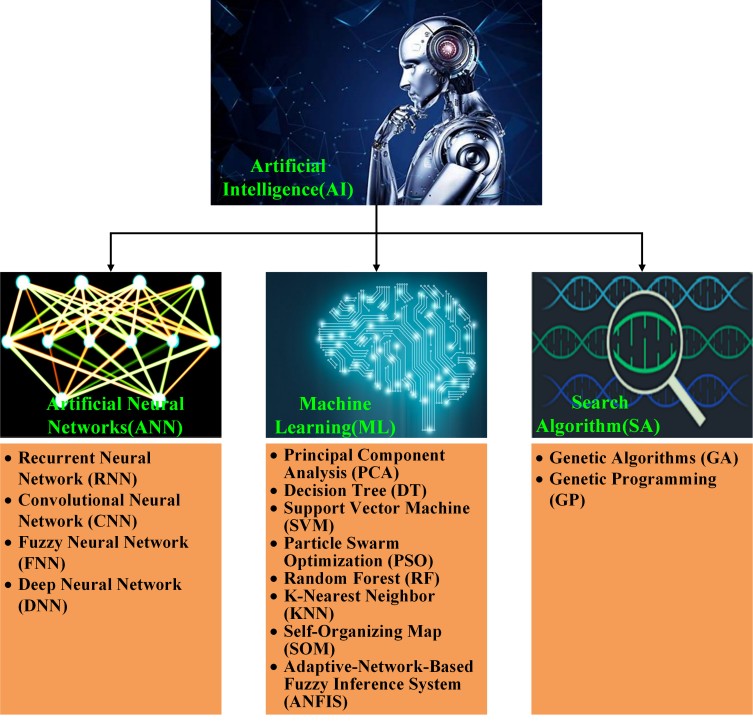

**Figure 2.** Classification of AI models for WWT.

## 2.1. Artificial Neural Network (ANN)

ANN is a mathematical model that imitates the behavioral characteristics of a biological neural network (NN) to process information. In ANN models, unit nodes are used to simulate neurons, and information processing is achieved by adjusting the weights of interconnection among a large number of nodes (neurons) in the neural network. Usually, ANN consists of an input layer, an output layer, and some hidden layers between the input and output layers. In ANN, many variable weights between neurons and active functions, such as sigmoid, tanh and ReLU functions, are used to perform complex nonlinear computation [19]. As the number of hidden layers of ANN increases, ANN can build more complex nonlinear models and its expression ability enhances. Thus, ANN can be trained to learn complex nonlinear relationships between inputs and outputs by constructing and optimizing loss functions [20]. The most commonly used ANN models mainly include Recurrent Neural Networks (RNNs), Convolutional Neural Networks (CNNs), Fuzzy Neural Networks (FNNs) and Deep Neural Networks (DNNs). The basic architectures of ANN, RNN, CNN, FNN and DNN models are shown in Figure 3, where Figure 3a displays an ANN architecture, including neurons represented by the circles; an input layer fed by input variables 1, 2, . . ., n; two hidden layers in the middle; and an output layer with output variables 1, 2, . . ., n. Next, we will present a brief introduction of these ANN models.

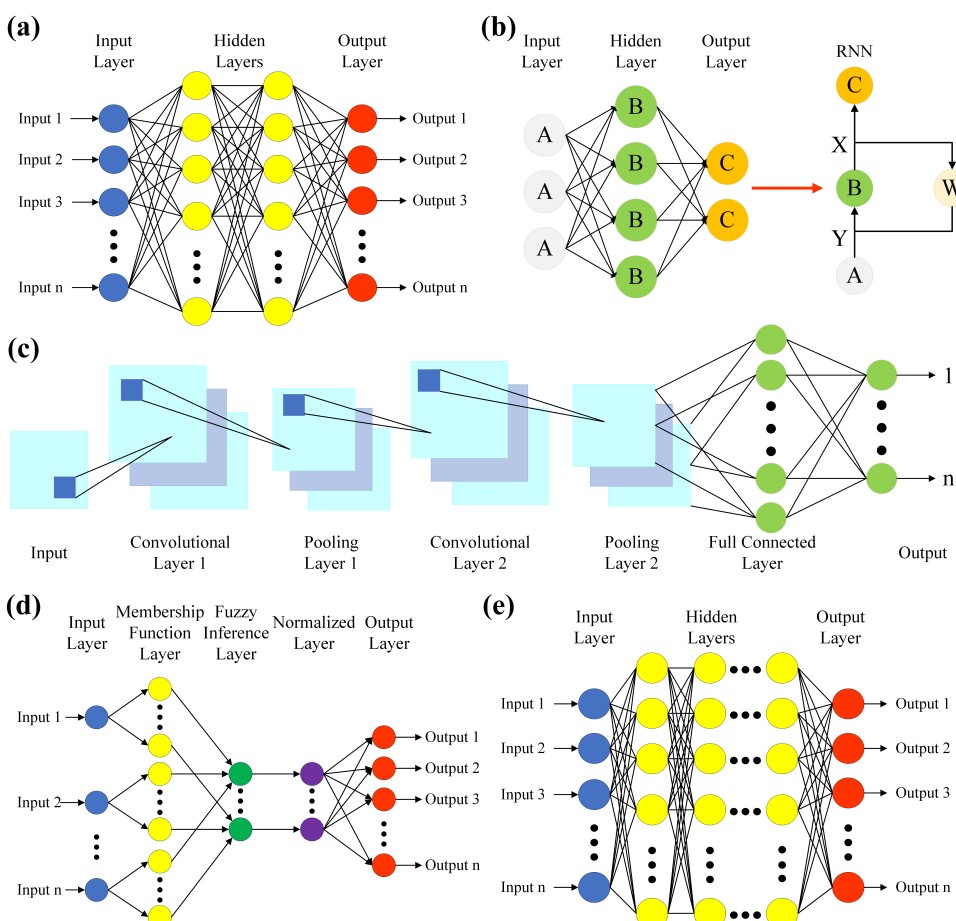

**Figure 3.** A basic architecture of ANN models. (**a**) ANN; (**b**) RNN; (**c**) CNN; (**d**) FNN; (**e**) DNN.

### 2.1.1. Recurrent Neural Network (RNN)

RNN is a class of NN with feedback connections that takes sequence data as inputs and makes recursion in the evolution direction of the sequence. RNN has the abilities of memory, parameter sharing and Turing completeness, so it has certain advantages in learning the nonlinear characteristics of time series problems. The most commonly used RNN is long short-term memory (LSTM), which solves the gradient disappearance problem

in traditional RNN by adding additional gated units [21]. RNN achieves great success in the applications of water and WWT, water quality management and water-based agriculture. A simple RNN architecture is shown in Figure 3b, where the output of the hidden layer is stored in the memory "W", which can be considered as another input in the RNN.

### 2.1.2. Convolutional Neural Network (CNN)

CNN is a class of Feedforward Neural Networks with convolutional computation and deep structure. It is one of the representative algorithms of deep learning (DL) with representation learning ability and has been widely used in computer vision, natural language processing and other fields. CNN extracts the complex features of input images through the convolutional layers, reduces the feature dimension through the pooling layers and, finally, realizes the task of classification or regression through the fully connected layers [22]. Figure 3c depicts a typical CNN architecture, which consists of input, output, convolutional, pooling and fully connected layers.

### 2.1.3. Fuzzy Neural Network (FNN)

FNN is a hybrid NN model that combines the advantages of fuzzy logic and ANN to handle problems with uncertainty or ambiguity. FNN uses fuzzy logic reasoning to process the input data and then applies ANN to train and output the results. FNN has a similar structure to traditional NN, but it uses fuzzy logic (membership function, fuzzy inference and normalization) to describe the fuzzy relationship between inputs and outputs, as well as the connection weights among neurons. A typical FNN architecture, including input, membership function, fuzzy inference, normalized and output layers, is shown in Figure 3d. FNN has some advantages in solving the problems that are difficult for traditional NN, and has been widely used in pattern recognition, control system, predictive analysis and so on.

### 2.1.4. Deep Neural Network (DNN)

DNN is a type of ANN with multiple hidden layers between the input and output layers. The deep architecture of DNN enables it to learn hierarchical representations of data, where higher-level features are learned by combining lower-level features in successive layers. Similar to other neural networks, DNN consists of more hidden layers and neurons and has been widely used for learning highly nonlinear mappings from inputs to outputs or capturing complex patterns in data. However, DNN needs a large amount of data to train because of the complex network architecture, which makes training difficult and computation expensive. Figure 3e displays a common DNN architecture with input, output and multiple hidden layers.

### 2.2. Machine Learning (ML)

ML is a subfield of AI that focuses on the development of algorithms and statistical models, and enables computer systems to automatically learn from data without being explicitly programmed [23]. The primary goal of ML is to build predictive models that can make accurate predictions or decisions based on what it has learned from data [24]. The most commonly used ML models include Principal Component Analysis (PCA), Decision Tree (DT), Support Vector Machine (SVM), Particle Swarm Optimization (PSO), Random Forest (RF), K-Nearest Neighbor (KNN), Self-Organizing Map (SOM) and Adaptive-Network-Based Fuzzy Inference System (ANFIS). Figures 4 and 5 present a schematic diagram of ML models, including PCA, DT, SVM and PSO and RF, SOM, KNN and ANFIS, respectively.

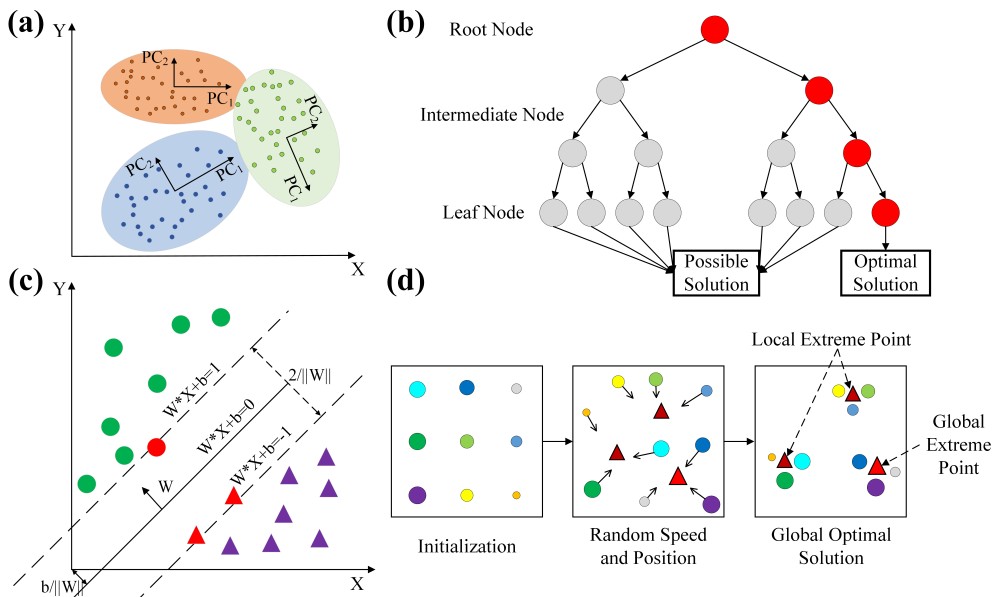

**Figure 4.** A schematic diagram of ML models. (**a**) PCA; (**b**) DT; (**c**) SVM; (**d**) PSO.

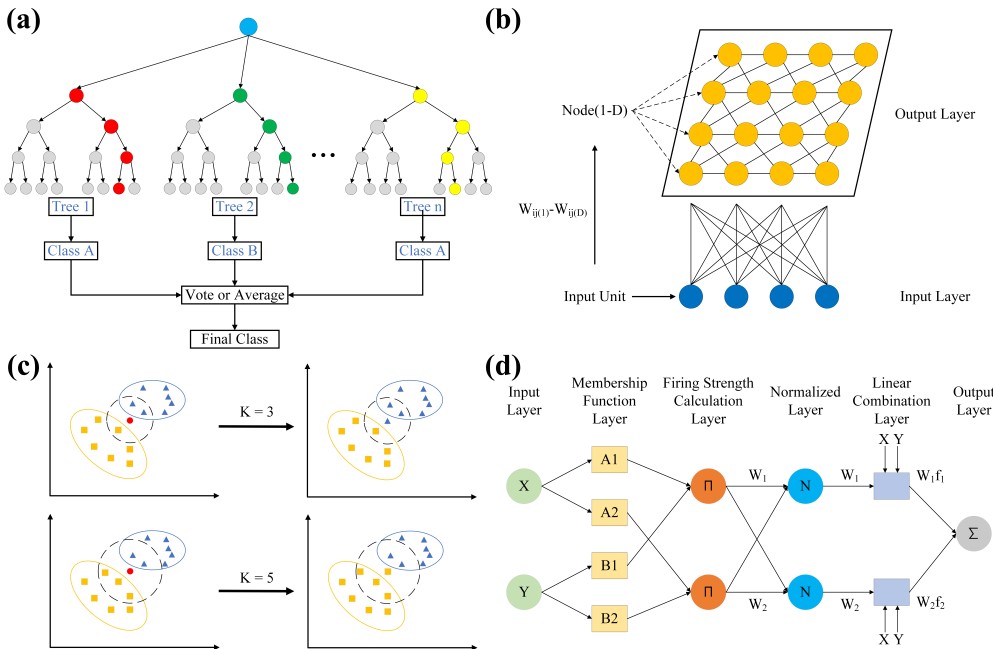

**Figure 5.** A schematic diagram of ML models. (**a**) RF; (**b**) SOM; (**c**) KNN; (**d**) ANFIS.

### 2.2.1. Principal Component Analysis (PCA)

PCA is a simple multivariate statistical machine learning algorithm and a commonly used data dimensionality reduction technique that can convert high-dimensional data to low-dimensional data while retaining most of the original data information by an orthogonal transformation [25]. It extracts some of the largest principal components based on the variances of variables for a better understanding of the system, which are orthogonal to each other. PCA is widely used in areas such as data clustering, image processing, natural language processing, noise filtering and other fields. A schematic diagram of PCA is presented in Figure 4a, where "$PC_1$" and "$PC_2$" denote the first and second principal components of datasets, respectively.

### 2.2.2. Decision Tree (DT)

DT is a common ML algorithm that can be used for classification and regression problems. DT divides the datasets into different subsets, and each subset is further divided according to the value of a certain feature. This process can be regarded as the construction of a tree with root, intermediate and leaf nodes, where each node represents a feature. Each branch represents the value of the feature, and every leaf node denotes the real classification or regression result [26]. DT is easy to understand and interpret, and can handle missing data, outliers and nonlinear relationships with high accuracy. It can also be used for feature selection, but it is easy to overfit, especially for higher-dimensional datasets. Figure 4b exhibits a typical DT architecture with branches and root, intermediate and leaf nodes.

### 2.2.3. Support Vector Machine (SVM)

SVM is a commonly used ML algorithm for classification and regression problems. It maps data to a higher dimensional feature space using kernel functions, such as polynomial kernel and radial basis kernel functions. It then searches an optimal hyperplane that separates different categories of data in the feature space based on the greatest distance from the nearest data point to the hyperplane [27]. SVM can deal with high-dimensional data and nonlinear relationships, and avoid the problem of local optimal solutions. However, it is inefficient for big datasets, sensitive to noise and outliers and may have difficulty in selecting suitable kernel functions and hyperparameters. Figure 4c displays an example of SVM for classification, where the solid line is a hyperplane acting as a decision boundary and the two parallel dashed lines represent spacing boundaries.

### 2.2.4. Particle Swarm Optimization (PSO)

PSO is a commonly used optimization algorithm for solving complex optimization problems. It imitates the swarm behavior of biological populations, such as birds or fishes, by constantly adjusting the individual position and velocity to search the optimal solutions. In each iteration of the algorithm, each individual represented by a particle updates its position and velocity and moves to the best-known position based on the optimal position in the population and the defined rule [28]. PSO can deal with various complex nonlinear problems and avoid falling into local optimal solutions, but it is sensitive to the initial conditions of the problem, resulting in the requirement of running many times to obtain good results. A schematic diagram of PSO to obtain the global optimal solution is shown in Figure 4d.

### 2.2.5. Random Forest (RF)

RF is an integrated ML algorithm and is mainly used for classification and regression problems. Its main idea is to improve the accuracy and stability of the model by integrating multiple DTs on data samples. In RF, to improve the diversity of the model, each DT is a basic classifier that is trained on a random subset, usually randomly drawn from the original datasets. The final classification or optimization is obtained by voting or averaging the results of multiple DTs [29]. RF can deal with high-dimensional data and complex nonlinear problems, and avoid overfitting. However, it may be sensitive to certain noises and outliers and requires more computing time and resources to train, especially under the condition of high tree density. Figure 5a depicts a schematic diagram of RF for classification by integrating n DTs.

### 2.2.6. Self-Organizing Map (SOM)

SOM is a commonly used ML algorithm for clustering and dimensionality reduction and only consists of input and output layers. It projects high-dimensional input data into a low-dimensional space and associates similar input data through a competitive process while preserving the topology of the data. SOM, as an unsupervised ML method, can reduce the dimensionality of complex datasets in a low-dimensional mapping space, which makes it easier to visualize and classify data points [30]. It is an effective visualization

method to help understand high-dimensional data and has been widely used in data visualization, clustering, classification and other fields owing to its simplicity and practicality. A schematic diagram of SOM is presented in Figure 5b.

### 2.2.7. K-Nearest Neighbor (KNN)

KNN is a simple and commonly used ML algorithm used for classification and regression. It predicts the label of a new data point according to the label of K neighbors closest to the new data point in the training datasets. KNN has the ability to deal with complex nonlinear relationships between inputs and outputs, adjust the complexity of the model adaptively and adapt to various data types using different distance measurements. However, it may be sensitive to high-dimensional datasets and noise, and computationally expensive for big datasets. Figure 5c displays an example of KNN before and after classification for the new data (solid red circle) when $K = 3$ and $K = 5$.

### 2.2.8. Adaptive-Network-Based Fuzzy Inference System (ANFIS)

ANFIS is a new hybrid intelligent inference system based on fuzzy logic and neural network for regression, classification and prediction. It effectively combines the fuzzy inference capacity of fuzzy logic with the learning ability of neural networks to realize the adaptive learning of fuzzy inference systems [31]. ANFIS and FNN have some similarities, but the network structure, computational method and application scope are different. A typical ANFIS architecture with input, membership function, firing strength calculation, normalized, linear combination and output layers is shown in Figure 5d. ANFIS can be used for modeling nonlinear and multivariable complex systems and has been widely applied in the areas of fuzzy control, prediction and classification.

### 2.3. Search Algorithm (SA)

Except for ANN and ML, SAs are also commonly used AI models, including Genetic Algorithm (GA) and Genetic Programming (GP), as shown in Figure 6. A brief introduction to GA and GP is given in the following.

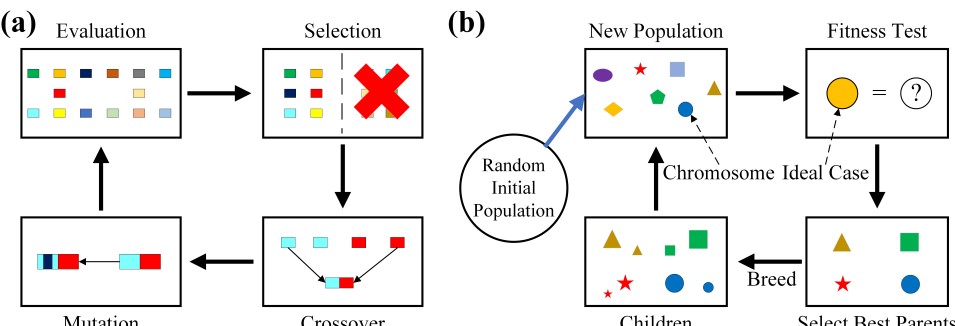

**Figure 6.** A schematic diagram of SA models. (**a**) GA; (**b**) GP.

### 2.3.1. Genetic Algorithm (GA)

GA is a commonly used search and optimization method based on the principle of biological evolution. It imitates the genetic process in nature and obtains excellent individuals better adapted to the environment through genetic manipulation. The genetic manipulation mainly includes selection, crossover and mutation, as depicted in Figure 6a. GA can search the optimal solution in multidimensional spaces and is not limited by the local optimal solution. Thus, it has good global search ability and strong adaptability and can deal with complex nonlinear optimization problems [32]. GA is widely applied in ML, AI, control systems, optimization design and other fields.

### 2.3.2. Genetic Programming (GP)

GP is an evolutionary computing technique based on GA that automates the generation and selection of computer programming inspired by biological evolutionary processes to perform regression, classification and optimization. Unlike other ML algorithms, GP is performed automatically by randomly generating an initial population and then using GA to evolve [33]. A schematic diagram of GP is displayed in Figure 6b. GP has good adaptability and generalization ability, and is often used to solve problems that are highly nonlinear or have no explicit analytic form, such as image recognition and prediction.

In fact, each AI model has advantages, disadvantages and application scopes. Table 1 lists the commonly used AI models for WWT, their purposes, advantages and disadvantages. An appropriate AI model should be selected carefully depending on its advantages and the characteristics of the problem to be solved, so as to get the best results. Meanwhile, Table 2 summarizes the commonly used activation functions of AI models for WWT. The expression and output range of every activation function are presented, and the frequency of use in the literature is marked with the blue symbol "*". The more symbols there are, the higher frequency of its use. Additionally, we present a general flow chart in Figure 7 to illustrate how AI models are applied in WWT. To evaluate the performance of AI models, some commonly used indicators are required, for example, mean squared error (MSE), root mean squared error (RMSE), sum of squared error (SSE), mean absolute error (MAE) and coefficient of determination ($R^2$). The definitions and details of these common indicators are omitted here; interested readers can refer to the literature [34].

**Table 1.** Commonly used AI models for WWT, their purposes, advantages and disadvantages.

| AI Models | Purposes | Advantages | Disadvantages | Ref. |
|---|---|---|---|---|
| RNN | Regression Classification Prediction | Suitable for time series modeling No limit to the length of inputs | Computationally expensive Training difficulty | [35] |
| CNN | Regression Classification Segmentation | Suitable for image-related modeling Extracting important features of images | Computationally expensive Training difficulty | [36] |
| FNN | Regression Classification Prediction | Easy to implement and interpret Suitable for complex nonlinear problems | Computationally expensive Complex model architecture | [37] |
| DNN | Regression Classification Prediction | Accurate and fast prediction Suitable for complex nonlinear problems | Computationally expensive Training difficulty Easy to overfit | [36] |
| PCA | Clustering | Simple and easy to implement Reduces dimensionality | May lose some important information Sensitive to noise data | [38] |
| DT | Regression Classification Optimization | Easy to understand, interpret and classify No need to preprocess | Low training efficiency Not suitable for imbalanced datasets | - |
| SVM | Regression Classification Prediction | Can handle high-dimensional problems Suitable for complex separable datasets | Computationally expensive Not suitable for larger datasets | [37] |
| PSO | Regression Classification Clustering | Simple and easy to use High computational efficiency Strong universality | Sensitive to initial conditions Not suitable for discrete problems | [39] |
| RF | Regression Classification Prediction | Simple and easy to use Suitable for high-dimensional datasets Strong generalization | Need dense decision trees to guarantee accuracy and robustness Computationally expensive | [36] |

**Table 1.** *Cont.*

| AI Models | Purposes | Advantages | Disadvantages | Ref. |
|---|---|---|---|---|
| KNN | Regression Classification | Simple and easy to use Suitable for nonlinear classification | Computationally expensive High memory consumption | [36] |
| SOM | Clustering | Suitable for high-dimensional datasets Reduces dimension | High computational complexity Not suitable for missing datasets | [40] |
| ANFIS | Regression Classification Prediction | Combine the advantages of ANN and FIS Use determination and fuzzy data | Computationally expensive Hard to define appropriate membership function | [37] |
| GA | Regression Classification Optimization | Suitable for complex nonlinear problems Support multi-objective optimization Efficient and flexible | Difficult to train Poor local search ability Not suitable for high dimensions | [39] |
| GP | Regression Classification Optimization | Suitable for complex optimization problems Optimize by automatic search | Many control variables Converge slowly Not suitable for high dimensions | - |

**Table 2.** Commonly used activation function of AI models for WWT. The blue symbol "∗" represents the frequency of use in the literature.

| Activation Function | Expression | Output Range | Ref. |
|---|---|---|---|
| Sigmoid *** | $f(x) = \frac{1}{1+e^{-x}}$ | $(0,1)$ | [41] |
| Tanh *** | $f(x) = \tanh(x)$ | $(-1,1)$ | [42] |
| ReLU ** | $f(x) = \max(0,x)$ | $[0,+\infty)$ | [36] |
| Leaky ReLU * | $f(x) = \max(\lambda x, x)$ | $(-\infty, +\infty)$ | [43] |
| ELU ** | $f(x) = \begin{cases} a(e^x - 1), x < 0 \\ x, x \geq 0 \end{cases}$ | $(-a, +\infty)$ | [44] |
| Heaviside * | $f(x) = \begin{cases} 0, x < T \\ 1, x \geq T \end{cases}$ | $[0,1]$ | [45,46] |
| Ramp * | $f(x) = \begin{cases} 0, x < T_1 \\ \frac{x-T_1}{T_2-T_1}, T_1 \leq x \leq T_2 \\ 1, x \geq T_2 \end{cases}$ | $[0,1]$ | [47] |
| Linear * | $f(x) = x$ | $(-\infty, +\infty)$ | [48] |

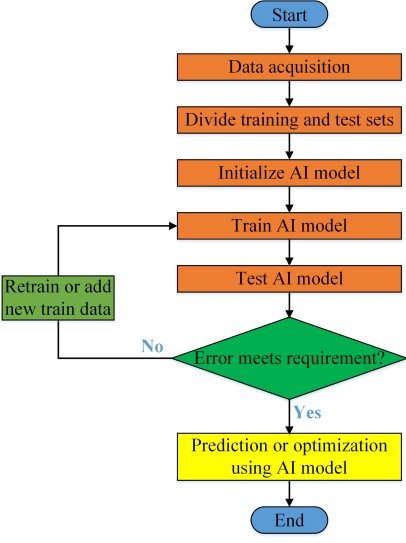

**Figure 7.** Flow chart for the applications of AI models in WWT.

## 2.4. Hybrid AI Models

Hybrid AI models can take full advantage of individual models to improve the prediction or optimization performance of AI models by integrating two or more of the above AI models. As shown in Table 1, every AI model has some drawbacks used for WWT. Hybrid AI models overcome the major disadvantages of a single AI model and, thus, show stronger learning and prediction abilities in dealing with more complex nonlinear problems. In the literature, GA, PSO, RNN and SVM are the commonly used AI models for combination with other AI models to obtain a more effective hybrid AI model with better performance, such as GA-SVR, GA-ANN, GA-FNN, PSO-RNN, PSO-SVM, PSO-ANN, ANN-GANN and SVM-SA [49]. Hybrid AI models have shown their great potential in solving new or difficult environmental problems related to sewage and are receiving increasing attention from researchers [50–52].

## 3. Applications of AI Models in WWT

AI models along with the Internet of Things (IoT) framework and conventional methods are helpful for the design of smart WWT systems and the reuse of sewage [53]. An AI model is a useful and powerful tool for the modeling, prediction and optimization of the WWT process and has been widely applied in various aspects of WWT, such as the removal of dyes, heavy metals, nutrients, organics, solids, microbial contamination, drugs and pesticides from water [49,54–56]. From the viewpoint of research scale, the applications of AI models are mainly in laboratory-scale research and process design. In practical applications, process design usually consists of process parameter optimization and process performance prediction. In optimization and prediction, a large amount of data is required to establish and train AI models, which can be achieved by monitoring water quality.

### 3.1. Water Quality Monitoring for Data Acquisition

In the applications of AI models in WWT, water quality monitoring is an important method to obtain water quality parameters or data. A lot of data are available by using sensors to continuously monitor influent and effluent water quality [34]. Although different numbers of datasets as inputs are employed for different studies, the percentage of data for training and testing AI models almost remains the same. Most studies use 60–80% data for training and the remaining data for testing. Some researchers have made an attempt to design various sensors to enable rapid and accurate real-time monitoring and WWT process automation by real-time sensing, data analysis and online controls [57–60]. For the monitoring of influent water, some water quality parameters, such as BOD, COD, pH, DO, flow rate, temperature and initial pollutant concentration, are easily obtained and used for the inputs of AI models, while for the monitoring of effluent water, some water quality parameters, such as effluent BOD, COD, pH, DO and pollutant concentration, are usually used to evaluate the effect of WWT or the performance of wastewater treatment plants (WWTPs).

An increasing number of measured data and AI models, as well as multivariate statistical methods, have made data-driven modeling and real-time prediction attractive. Post et al. [61] combined a CNN model with laser-induced Raman and fluorescence spectroscopy (LIRFS) to achieve real-time monitoring of the micropollutants of WWTP with a correlation coefficient of $R^2 = 0.74$ for all samples. The results show that this method can lead to high-precision measurement results, reach detection limits and detect micropollutants that cannot be monitored using the monitoring methods of WWTPs. The combination of sensitive fluorescence measurements with very specific Raman measurements supplemented with AI is a promising real-time monitoring tool for image recognition in WWT-related fields, such as microbiological water quality tests, micropollutants identifications and even device-specific adjustments in WWTP management. Based on PCA, ANN and multivariate statistical process control, Lee et al. [38] developed a real-time remote monitoring system for WWTP to monitor operating statuses and provided the key information needed for efficient operation from the experts. Mustafa et al. [62] reviewed

the applications of IoT and AI models in water quality monitoring and prediction with high accuracy, which can provide safe water quality services for users and an important basis for government water quality management decision making. For water quality monitoring using soft sensors, a literature review was performed by Haimi et al. [63] to summarize the applications of data-derived soft sensors for the monitoring and online prediction of biological WWTP. Ching, So and Morck [64] also conducted a systematic review of advances in soft sensors for the online monitoring of WWTP. Schneider et al. [65] performed an experimental study to identify sensors to monitor on-site WWTP without sensor maintenance over one year and showed that robust soft sensors can be reasonably designed to meet real-time monitoring tasks while reducing the maintenance frequency dramatically.

*3.2. Laboratory-Scale Research*

Owing to low cost, short experiment period, security controllability, easy operation and good repeatability, laboratory-scale research has been an important tool for developing new process designs and WWT technologies. We conclude the common applications of AI models for laboratory-scale research in Table 3. In laboratory-scale research, the applications of AI models in WWT mainly focus on the design and optimization of membrane processes or bioreactors. Some researchers have made comprehensive reviews of the applications of AI models for WWT using membrane processes or membrane bioreactors (MBRs) in recent studies [66–68]. MBR is an efficient and useful wastewater treatment process combining biological (microbial) treatment with membrane filtration. It is a hybrid system essentially that integrates a conventional biological treatment system and critical physical liquid–solid separation functions achieved by membrane filtration equipment. Because MBR combines the technical superiorities of membrane-based physical separation and microorganism-based biodegradation, it has several advantages over the traditional activated sludge process, such as higher biomass concentration, less sludge production, shorter hydraulic retention time (HRT), eliminating the need for secondary clarifiers, smaller plant space requirement and improved effluent quality [69]. Thus, as one of the most important innovative technologies in WWT, MBR is an efficient tool for sustainable wastewater management and has been widely used for the treatment of various municipal and industrial wastewater. Recently, Rahman et al. [70] performed a review of the historical advancement in MBR technology toward sustainable wastewater management. Tomczak and Gryta [71] outlined energy-efficient anaerobic MBR (AnMBR) technology for WWT and demonstrated that AnMBRs have lower energy demand than typical WWTPs.

A variety of anaerobic [72] and aerobic [73] MBRs have been developed and applied in WWT, but a main barrier for the widespread application of membrane processes or MBRs is membrane fouling since it significantly decreases the performance and lifespan of membranes, resulting in increased maintenance and operating costs. Membrane fouling induces a decrease in the permeability of membranes, and is a complex and inevitable phenomenon, which is attributed to the accumulation and adsorption of the pollutants in wastewater on the membrane surface and inside the membrane pores. Overall, membrane fouling is the key impact factor limiting the performance and cost of membrane processes, and researchers have been trying to develop efficient and sustainable fouling control strategies. Tomczak, Grubecki and Gryta [74] proposed a method for membrane fouling control in an MBR using 1% NaOH solutions and demonstrated that the method is effective in restoring the initial membrane performance. In the membrane processes or MBR-related application studies, how to control membrane fouling and optimize MBR performance effectively and economically are two of the most central questions for the rapid commercialization of and large-scale applications in WWT [75–77].

For the applications of AI models to predict or control membrane fouling, the commonly studied membrane types are forward osmosis, reverse osmosis, nanofiltration, ultrafiltration and microfiltration [78]. Chen et al. [79] applied a radial basis function (RBF) ANN model to quantify interfacial energy with a randomly rough membrane surface in the membrane fouling process. They showed that the RBF-ANN model can well capture the complex relationships between interfacial energy and key influencing factors. Jawad et al. [80] adopted an ANN model to predict permeate flux for a lab-scale forward osmosis process with high accuracy, $R^2 = 0.973$. The results indicate that the ANN model performs better than the MLP model, and a lower number of hidden layers and a higher number of neurons are helpful to improve the accuracy of the ANN model. Subsequently, they presented a hybrid ANN-RSM model to further simulate the forward osmosis process and predict membrane flux [81]. In the developed hybrid model, the ANN model predicting the membrane flux is used for the experimental design, while the RSM model is used for optimization. The prediction performance for the ANN and RSM models are $R^2 = 0.98036$ and 0.9408, respectively.

For the applications of AI models to optimize MBR performance, various operating parameters, such as temperature, pH, DO, salinity, HRT, pollution load, BOD, COD and pollutant concentration, are considered to determine the optimum processing condition [82]. Zaghloul et al. [83] proposed a five-stage ML model to simulate and predict the behaviors of aerobic granular sludge (AGS) reactors using 475 days of data collected from three lab-based reactors and adopted an ensemble of ANN, SVR and ANFIS models to improve the predictive performance. They found the model can forecast the behaviors of AGS reactors with average $R^2 = 95.7\%$, RMSE = 0.032 and MAPE = 3.7%. Ren et al. [84] used a Backpropagation Neural Network (BPNN) model to simulate the removal of $COD_filt$ and COD by conducting a pilot-scale submerged MBR to treat high-strength Chinese traditional medicine wastewater and confirmed that the model can accurately predict the removal rates and help to obtain the optimum operational conditions. Cai et al. [85] conducted an aerobic–anaerobic micro-sludge MBR (O-AMSMBR) to study the effect of pH on pollutant removal performance of a reactor using Wavelet Neural Network (WNN) and BPNN models and showed that pH is a key factor affecting the COD and TN removal efficiencies of O-AMSMBR. To optimize the processing efficiency of MBRs, they further studied the effects of various ecological factors on effluent marine domestic sewage by implementing an air-lift multilevel circulation MBR and analyzed their impacts on the O-AMSMBR performance using the BPNN model [86]. The results show that the order of relative importance for the ecological factors is pH ≈ MLSS > HRT > COD, which indicates that pH is significant and should be considered in implementing AI models to evaluate the effectiveness of MBR systems.

**Table 3.** Applications of AI models for laboratory-scale research.

| AI Model Used | Input Variables | Output Variables | Remarks | Reactor Type | Ref. |
|---|---|---|---|---|---|
| ANN, ANFIS, SVM | Influent $NH_4 - N$, $PO_4^{3-}$, pH, OLR, HRT, etc. | Effluent COD, $NH_4 - N$ and $PO_4^{3-}$ | New multi-stage ML model for better prediction of AGS reactor performance. An ensemble of ML for more accurate predictions. | AGS reactor | [83] |
| ANN | COD, MLSS, MLVSS, pH, DO, Alkalinity, TN, TP, $NO_3^- - N$, $NH_4 - N$ | Transmembrane pressure (TMP) | New ANN model to accurately predict membrane fouling. Identify an optimal parameter set to predict TMP using ANN. | Anoxic–aerobic MBR | [87] |
| ANN, ANFIS | Influent COD, pH, oil and grease removal, etc. | Biogas production | New ANN and ANFIS models to predict biogas production from spearmint essential oil WWT. Obtain the best BP-ANN and ANFIS topologies. | UASB | [88] |
| ANN | HRT, temperature, composition and chemical dose | Methane production | ANN model to forecast biogas production and identify the optimum process conditions. Chemical treatment enhances anaerobic digestion efficiency. | Anaerobic reactor | [89] |
| ANN | Volatile solid, pH, organic load rate, HRT, temperature, reactor volume | Biogas production | ANN model to predict biogas production from food, fruits and vegetable wastes. Assess different ANN topologies and build database. | Anaerobic reactor | [90] |
| ANN | MLSS concentration, HRT and time | TMP and COD removal percentage | ANN model to simulate and predict TMP and COD removal percentage of MBR. Protein in biofilm/cake EPS is the dominant fouling factor. | MBR | [91] |
| PCA, fuzzy clustering (FC) | TMP | Principal components of TMP | PCA and FC to assess membrane fouling. PCA-FC model for membrane fouling control. | MBR | [92] |

**Table 3.** *Cont.*

| AI Model Used | Input Variables | Output Variables | Remarks | Reactor Type | Ref. |
|---|---|---|---|---|---|
| ANN | Flux, aeration ratio, initial TMP, operating time, etc. | TMP | Mathematical and ANN models to predict membrane fouling. Mathematical model has a better stability and ANN has a better prediction performance. | Intermittently aerated MBR | [93] |
| Recurrent fuzzy NN | Influent COD, $NH_4 - N$, pH, BOD, SS, TP, etc. | Membrane permeability | Intelligent detecting system to evaluate MBR performance. Suitable for online detecting membrane fouling. | MBR | [94] |
| MLP, ANN, GA | Time, TSS, influent COD, SRT, MLSS | TMP or permeability | GA-ANN model to evaluate membrane fouling. GA-ANN predicts TMP and permeability accurately. | Submerged MBR | [95] |
| ANN | Influent concentrations of COD, $NH_4 - N$ and TN, etc. | Effluent concentrations of COD, $NH_4 - N$, TN | New ANN model to predict biofilm system performance. The new model performs the best. | Biofilm system | [96] |
| WNN, BPNN | pH, sludge loading, salinity, COD or TN volume loading rate | Effluent concentrations of COD or TN | WNN and BPNN models to study the effect of pH on pollutant removal. pH is the key factor for biodegradation. | $O - AMSMBR$ | [85] |
| BPNN | Influent concentrations of COD, BOD, etc. | Effluent concentrations of COD, etc. | New BPNN model to simulate AnMBR performance. AnMBR can treat pharmaceutical wastewater efficiently. | AnMBR | [97] |
| ANN, GA | Conductivity, organic loading rate, temperature | COD removal efficiency | New ANN-GA model to predict and optimize COD removal efficiency. ANN-GA improves reactor performance. | UASB | [98] |

### 3.3. Process Design

WWT process involves complex process design and operating conditions, and AI models have shown great advantages in minimizing or reducing the complexities of the WWT process. AI models can effectively and easily establish a complex relationship between the various input and output variables. The commonly used input variables are time, temperature, pH, initial concentration of pollutants and influent water quality parameters, and the output variables are mainly the removal efficiency and the adsorption efficiency of contaminants or effluent water quality parameters. AI models have been successfully applied in various aspects of WWT, such as the prediction of effluent water quality and WWT performance, as well as the optimization of energy consumption and operating parameters [96]. From the perspective of process design of WWT, the applications of AI models are concentrated in the optimization of process parameters and the prediction of process performance [99].

#### 3.3.1. Process Parameters Optimization

The main purpose of process parameter optimization is to reduce costs and increase the efficiency of WWT. Table 4 summarizes the applications of AI models for the optimization of process parameters. Nayak et al. [100] used a hybrid ANN-GA model to predict the optimal process conditions for enhancing the biomass of the green microalga in an algal biorefinery. They found 4-12-1 topology is the optimal network architecture with a maximum correlation coefficient $R = 0.9947$ and minimum MSE, and these parameters improved the algal biomass productivity by about 57% and had a $CO_2$ sequestration rate of $578.1 \pm 23.1 \, \text{mg} \, \text{L}^{-1} \, \text{d}^{-1}$ and a COD reduction of $95.9 \pm 2.4\%$. Qi et al. [39] applied RSM, ANN-PSO and ANN-GA models to study the decontamination of methylene blue (MB) from simulated wastewater using mesoporous rGO/Fe/Co nanohybrids. The results show that the ANN-PSO model has the best performance among these models in the prediction of the optimum conditions for decontamination efficiency. The mesoporous nanohybrids could be used as a low-cost and fast decontaminant material to treat organic contaminants or other pollutants in wastewater. Martín de la Vega and Jaramillo-Morán [40] adopted SOM to identify four key parameters of a municipal WWTP running by monitoring Oxidation–Reduction Potential (ORP) and DO based on three thousand two hundred aeration–non-aeration cycles. This method can improve the removal efficiency of nutrients in WWTP.

For process parameter optimization, Picos-Benítez et al. [101] utilized an ANN-GA model to predict the treatment performance of sulfate wastewaters with Bromophenol blue dye using an electro-oxidation (EO) process and obtained the optimum operational conditions. They found that the AI model is a powerful tool in designing and controlling the WWT processes. For MB WWT, ANN and ANFIS models were employed by Aghilesh et al. [102] to obtain the optimum conditions for MB removal using low-cost agricultural waste (sugarcane bagasse and peanut hulls). They also performed Fourier Transform Infra-Red (FTIR) spectral analysis to confirm the biosorption and the distinguished prediction performance of these AI models for biosorption. In fact, FTIR is a useful spectroscopic technique and has been widely used for the detection and analysis of pollutants, such as microplastics in water [103], table salts [104] and nitrates from agricultural fertilizers in soil [105,106]. FTIR spectra can be used to analyze the chemical composition of pollutants and identify functional groups, providing important information about them present in compounds, complex substances and bio-sorbent surfaces, but spectra interpretation is time-consuming. To reduce the time to analyze functional groups so as to facilitate the interpretation of FTIR spectra, Enders et al. [107] developed the first generalizable model based on CNN to identify functional groups in gas-phase FTIR spectra. The results demonstrate that CNN models are effective at identifying spectral features and can be extended to other micropollutants or chemical identification application fields with a lot of spectral examples. Overall, image-based AI models coupled with spectroscopy techniques,

such as FTIR, SEM and Raman spectroscopy, are useful for the identification and detection of contaminants, especially micropollutants in water-related fields.

For a WWT system, Li et al. [43] proposed a hybrid deep leaning CLSTMA model based on CNN, LSTM and AM to monitor and model the water quality of a paper industrial WWT system for cleaner production. Compared with CNN, LSTM and CLSTM models, the CLSTMA model has better performance in monitoring and modeling water quality. An intelligent WWT system based on AI models and sensors was presented by Miao et al. [41] to assist in managing sewage treatment. A Gated Recurrent Unit (GRU) model performs better than LSTM and SVR models, and the intelligent WWT system can be extended to small-scale sewage industries in sustainable cities. Rodríguez-Rangel et al. [36] also explored five AI models to simulate and predict the biomass production of carbohydrates in WWT systems, considering the interactions of nutrients, carbon, biomass growth and population. The results indicate that the CNN-1D model has better performance than other models and can approximate system dynamics. For the study of WWTP, Hwangbo et al. [42] used DNN and LSTM to predict the $N_2O$ emission rate and identify the key parameters affecting the characteristics of $N_2O$ high emission. They found that the LSTM model performs better than the DNN model, and a hybrid model combining mechanistic with DL models is helpful in quantitatively describing and understanding complex $N_2O$ emission dynamics from WWTPs. Zhu, Jiang and Feng [37] also proposed an upgraded feedforward NN with the least square SVM (FFNN-LSSVM) method to forecast the effluent BOD/NH3-N of a WWTP. The proposed model has high predictive accuracy, limited computation duration and a simple calculation mechanism, and performs better than existing techniques in wastewater quality prediction.

**Table 4.** Applications of AI models for the optimization of process parameters.

| AI Models Used | Input Variables | Output Variables | Remarks | Pollution Type | Ref. |
|---|---|---|---|---|---|
| ANN, GA | Light intensity, photoperiod, temperature and initial pH | Biomass productivity | New ANN-GA model to predict optimal process conditions of an algal biorefinery. Productivity improved by 57%. | Green microalga | [100] |
| ANN, GA, PSO | Initial MB concentration, temperature, pH and contact time | Decontamination efficiency | ANN-PSO model to predict the optimum process conditions. rGO/Fe/Co nanohybrids can treat organic contaminants effectively. | MB | [39] |
| SOM | COD, BOD, TSS, TN, TP, etc. | Organic overload, working conditions | SOM model to optimize working conditions. Obtain key parameters and working conditions of biological nutrient removal. | Biological nutrient | [40] |
| ANN, GA | Electrolysis time, flow, current density, pH, dye concentration | Discoloration efficiency | ANN-GA model to optimize process conditions. AI can design, control and operate EO process. | Dye | [101] |
| ANN, ANFIS | Temperature, pH, bio-sorbent and dye concentration | Removal efficiency of MB | AI model to predict biosorption and obtain optimum conditions. Agricultural waste for effective biosorption of textile wastewater. | MB | [102] |
| CNN, LSTM, AM | Influent COD, SS, flux, DO, pH and temperature | Effluent COD and SS | New hybrid CLSTMA model to monitor water quality for cleaner production with low cost. | Paper wastewater | [43] |
| SVR, LSTM, GRU | Inflow and outflow COD temperature | Outflow COD | Intelligent WWT system based on ML and sensors. Applied it to a fine chemical plant. | City sewage | [41] |

**Table 4.** *Cont.*

| AI Models Used | Input Variables | Output Variables | Remarks | Pollution Type | Ref. |
|---|---|---|---|---|---|
| DNN, LSTM | DO, $NO_3^- - N$, $NH_4^+ - N$, influent and air flow rates, temperature | $N_2O$ concentration | Integrating mechanistic and DL models is very useful for understanding $N_2O$ emission dynamics. | $N_2O$ | [42] |
| ANN, CNN, LSTM KNN, RF | Mixed liquor, biomass, green algae, etc. | Carbohydrate content | Used 5 AI models to forecast biomass production. CNN-1D model performs the best. | Carbohydrate | [36] |
| ANN, SVM, FNN | Influent water quality, flow rate, etc. | Effluent BOD/NH3-N | New FFNN-LSSVM model to forecast water quality and optimize process parameters. | NH3-N nitrogen | [37] |

### 3.3.2. Process Performance Prediction

Process performance is an important aspect of the WWT process and has been the focus of researchers' attention. The common process performance predictions include the removal efficiency of pollutants, WWTP performance, optimal process condition and effluent quality. A brief summary of the applications of AI models for the prediction of process performance is presented in Table 5. For the prediction of WWTP performance, Nourani et al. [108] analyzed Nicosia WWTP performance using single AI and ensemble models. The results show that the ANFIS model performs better than other single AI models, and the NN ensemble model has the best prediction performance among the ensemble models. Xie et al. [109] also combined improved Feedforward Neural Network (IFFNN) with GA to predict the real-time effluent water quality of a WWTP in Jiangsu Province, and found the IFFNN-GA model enhances prediction performance by 52.3% (COD) and 72.6% (TN) compared with the traditional FFNN model. Deep cascade-forward backpropagation (DCB) and DL time series forecasting (DLTSF) models were presented by El-Rawy et al. [110] to predict the effluent water quality of the El-Berka WWTP and evaluate its treatment performance.

For the removal of contaminants, Bisaria et al. [111] employed ANN and ANFIS models to simulate the adsorption process of chlorpyrifos (CPS) using *Trapa bispinosa* peel (UFBC) and validated the effectiveness of adsorption experimentally. The results illustrate that UFBC is a sustainable and effective adsorbent with low equilibrium time and high adsorption capacity for CPS removal. For the prediction of effluent quality, Yang et al. [112] proposed a dynamic PCA-NARX model to predict the effluent quality and made potential real-time adjustments for WWTP operations. The results show that the dynamic model has better performance than static ANN models in modeling effluent quality. Nnaji et al. [113] predicted COD and CTSS removal efficiencies from textile wastewater using complex salt–Luffa cylindrica seed extract (CS-LCSE) as a coagulant based on RSM, ANN and ANFIS models. The results demonstrate that the ANFIS model has the best predictive performance with a higher $R^2$ value (0.9997 and 0.9996 for CTSS and COD removals) and a lower MSE value (0.0002643 and 0.0038472 for CTSS and COD removals). For the prediction of optimal process conditions, Mahmoud et al. [114] employed an ANN model to predict the COD removal efficiency from domestic wastewater by preparing Fe/Cu NPs under different operating conditions. The adsorption isotherm, kinetic studies and RSM results indicate that Fe/Cu NPs are an effective adsorbent material for COD removal, and the ANN model is useful to explore the optimum removal condition.

More and more studies have demonstrated that the utilization of AI in WWTPs can significantly enhance WWT efficiency and decision making, decrease environmental impacts and improve their performance (see Table 5) because AI algorithms can optimize various processes and operational conditions, such as temperature, pH, COD, chemical dosage, flow control and energy consumption. This leads to more efficient resource utilization, reduced operational costs and energy consumption, and improved overall system performance.

Additionally, a combination of AI technologies with the use of sensors can realize real-time monitoring and control, resulting in decreased response time, system failure risks and human resource costs. Overall, WWTPs that utilize AI technologies have shown superior performance compared to those without AI. The incorporation of AI enables enhanced efficiency, real-time monitoring, predictive maintenance, data-driven decision making and reduced environmental impact. Implementing AI in WWTPs can lead to more sustainable and effective management of our water resources.

Besides the applications of AI models in the prediction of process performance and the optimization of process parameters, AI models can be used in the pretreatment of wastewater to improve the pretreatment accuracy and the adaptive control accuracy of the system [115,116]. Some researchers have employed various AI models and their variants to comprehensively evaluate the water quality of rivers [117], lakes [118] and reservoirs [119].

**Table 5.** Applications of AI models for the prediction of process performance.

| AI Models Used | Input Variables | Output Variables | Remarks | Country | Ref. |
|---|---|---|---|---|---|
| FFNN, ANFIS, SVM | Influent pH, BOD, COD, conductivity, and TN | Effluent BOD, COD and TN | Used 3 AI models to predict Nicosia WWTP performance. NN ensemble model is more robust and reliable. | Cyprus | [108] |
| IFFNN, GA | Influent water quality, flow rate, etc. | Effluent water quality prediction at time $t + 1$ | New IFFNN-GA model to enhance real-time prediction of WWTP effluent quality. | China | [109] |
| FFNN, LSTM | Influent TSS, BOD, COD, ammonia and sulfide | Effluent TSS, BOD, COD, ammonia and sulfide | Different AI models to predict effluents and performance of WWTP. Recommend DCB and DLTSF models for evaluation and prediction. | Egypt | [110] |
| ANN, ANFIS | Contact time, adsorbent dose, pH, etc. | Adsorption efficiency | ANN and ANFIS models to predict the adsorption capacity of CPS by UFBC. New material is a sustainable and effective adsorbent. | India | [111] |
| PCA, NARX NN, ANN | pH, COD, BOD, TN, TP, SS, $NH_4^+$ and chromaticity | Effluent COD and TN | New PCA-NARX model to predict effluent water. Dynamic model performs better than static model. | China | [112] |
| ANN, ANFIS | pH, dosage and stirring time | COD and CTSS removal efficiencies | ANFIS model outperforms over ANN and RSM models. | Nigeria | [113] |
| ANN | pH, NP dose, contact time, etc. | COD removal efficiency | New ANN model to predict COD removal efficiency. Fe/Cu NPs are strong absorbents. | - | [114] |
| ELM, GA, SSA, PSO | Flow rate, temperature, pH, $NH_4$, EC, COD at time $t - 1$ | COD at time $t$ | KELM-SSA model performs better than other AI modes in predicting real-time water quality due to the combination of SSA. | Iran | [120] |
| ANN, PCA | Type 0041, Gordonia spp., etc. | Sludge volume index (SVI) | ANN and multivariate statistics to predict sludge volume index and assess filamentous bacteria. | South Africa | [121] |
| WNN | Influent COD, $NH_4^+ - N$, salinity | Effluent COD, $NH_4^+ - N$ | New WNN model can accurately evaluate the effect of salinity and predict pollutant removal processes. | China | [122] |
| WNN | TN and COD loading rates, HRT, pH | Effluent COD, TN | WNN model can forecast COD and TN removals and help long-term stable operation of WWTP. | China | [123] |

**Table 5.** *Cont.*

| AI Models Used | Input Variables | Output Variables | Remarks | Country | Ref. |
|---|---|---|---|---|---|
| ANN, GA | Time, OLR, RT, TDS | COD, TOC, MLSS, oil in sludge | ANN-GA model can predict hypersaline oily WWT processes and evaluate MBR performance. | Malaysia | [124] |
| FFNN, ANFIS, SVR | Influent BOD, COD, TSS and pH, etc. | Effluent BOD and COD at time *t* | AI models to predict WWTP effluent parameters. Using jittering and ensemble models simultaneously increases prediction accuracy. | Iran | [125] |
| SVM, ANFIS | Influent pH, TS, COD, etc. | Effluent Kjeldahl Nitrogen concentration | SVM and ANFIS models to predict removal efficiency of Kjeldahl Nitrogen. SVM can evaluate WWTP efficiency. | India | [126] |
| ANN | pH, adsorbent dose, contact time, etc. | Color removal efficiency | ANN and other models to simulate adsorption processes. GT-nZVI has a strong color removal ability for textile wastewater. | Egypt | [127] |

## 4. Challenges and Future Perspectives

Although AI models have many advantages over traditional models and have achieved great success in WWT-related fields demonstrated by all of the aforementioned studies, their disadvantages and limitations hindering widespread applications in WWT should not be ignored. The challenges and future perspectives of AI applications in WWT are summarized as follows:

1. An AI model such as ANN can describe complex nonlinear relationships between multiple inputs and multiple outputs, but it is a black-box and data-driven model essentially. That is to say, the AI model just offers a mapping relationship between inputs and outputs, but it cannot provide any mechanism information about the problem to be studied. AI models have proven to be a powerful tool and show good prospects in engineering applications for WWT fields; however, they have a long way to go in scientific research. The main reason is that the underlying mechanisms behind many WWT-related issues in the current research are still not clear. Although the AI models used in the above studies show good performances in solving specific problems and usually exhibit problem dependence, whether they can be applied to other WWT-related problems and their application scopes in WWT need to be studied. More importantly, traditional mathematical models [128], such as known knowledge, principles and equations, are called white-box models, which elucidate the underlying mechanisms, but have difficulty describing the complex nonlinear relationships between inputs and outputs of AI models. Combining an AI model (data-driven model) with a traditional mathematical model (knowledge-driven model) can dramatically reduce data requirements and allow for easily obtaining meaningful results [129]. The integrated black-box with white-box model is a promising tool for the study of the underlying mechanisms of WWT systems.

2. AI models usually have low interpretation because their parameters, such as neurons, hidden layers, weights and biases, have no physical significance, resulting in low interpretation. Another drawback of an AI model is poor reproducibility because of the random weights and biases [67]. Additionally, training NN, especially DNN, is difficult; thus, it is hard to obtain the optimal network parameters. An improperly trained NN may converge to a local minimum. Generally, NN provides different solutions under the conditions of different network architectures and parameters. There is no standard way to determine the best network architecture so far, which is often problem-dependent. Trial and error seems to be the only way, but this easily leads to overfitting or underfitting. For a particular real problem, the appropriate

selection of AI models, inputs, outputs, model architecture and datasets is vital for the results. How to construct AI models reasonably needs to be further studied. Moreover, more theoretical studies on AI techniques are needed to mitigate the difficulties of NN training, parameter optimization, poor reproducibility and low interpretation, so as to promote the development of AI applications in WWT.

3. AI models are heavily dependent on data, and big data are required in the training or learning process to guarantee prediction or optimization accuracy. In the WWT applications mentioned above, AI models are applied to different WWT-related fields and a lot of data with different formats and types have been collected, resulting in poor data management and difficulty in reuse by other researchers. Moreover, the data and source code used in the studies are rarely made public for various reasons, which leads to a lack of academic transparency. This is another reason why it is difficult to reproduce the results in the literature. Furthermore, researchers have used some statistical methods to evaluate the accuracy or precision of their AI models in almost every paper reviewed in this journal, such as $R^2$, MAE, MSE, RMSE, MAPE and SSE. The absence of open source code and data makes it difficult to fairly compare the performance of different AI models [12]. Due to the lack of benchmarks, standardization and fair comparison, it is hard for researchers to judge which AI model performs better for a specific real problem. Therefore, in order to reduce experiment costs, achieve fair comparisons and promote the widespread application of AI models in WWT, raw data and source code are encouraged to be made public and shared, and benchmark and standardization should be established.

## 5. Conclusions

This review summarizes the commonly used AI models and their applications in WWT, ranging from water quality monitoring, laboratory-scale research to process design. AI models are becoming more and more popular in WWT-related fields because of their strong learning and accurate prediction abilities. They have been successfully applied to model WWT systems, optimize process parameters, predict process performance and identify and detect contamination. Although AI models have many advantages and have become very useful tools for the treatment of wastewater, their disadvantages and limitations should not be ignored. Big data demand; poor data management; low interpretability; poor model reproducibility; and a lack of physical significance, mechanism explanation, academic transparency, standardization and fair comparison are important obstacles to the AI applications in relevant areas of WWT.

In order to overcome these hurdles and successfully apply AI models to WWT, mathematicians, biologists, engineers and computer experts should cooperate and develop new models or innovative technologies to design optimal WWT systems. Additionally, more studies from lab to field scales are needed to understand the complex behaviors of WWT systems with varying effect factors and to explore the mechanisms of key problems involved in WWT. Furthermore, hybrid AI models that integrate the advantages of two or more AI models and the newly emerging attention-based AI models could be solutions to complex water-treatment-related problems. The fusion of data-driven and knowledge-driven AI models is a new and promising method that is receiving increasing attention in WWT-related fields.

**Author Contributions:** Conceptualization, Y.W., Y.C. and D.L.; methodology, Y.W., Y.C., H.L., Q.G., C.D. and D.L.; validation, Y.W., Y.C., H.L., Q.G. and D.L.; formal analysis, Y.W., Y.C., H.L., Q.G., C.D. and D.L.; investigation, Y.W., Y.C., M.Z. and D.L.; writing—original draft preparation, Y.W. and Y.C.; writing—review and editing, Y.W., Y.C., C.D., M.Z. and D.L.; visualization, Y.W. and Y.C.; supervision, Y.W., M.Z. and D.L.; project administration, Y.W., M.Z. and D.L.; funding acquisition, Y.W., M.Z. and D.L. All authors have read and agreed to the published version of the manuscript.

**Funding:** This research was funded by the Key Research and Development Program of Zhejiang Province (grant No.: 2022C02045), the National Natural Science Foundation of China (grant No.: 12202319) and the National Key Research and Development Program of China (grant No.: 2018YFE0103700).

**Institutional Review Board Statement:** Not applicable.

**Informed Consent Statement:** Not applicable.

**Data Availability Statement:** Not applicable.

**Acknowledgments:** The authors sincerely thank the reviewers for their constructive comments to improve the manuscript.

**Conflicts of Interest:** The authors declare no conflict of interest.

## Abbreviations

The following abbreviations are used in this manuscript:

| | |
|---|---|
| AI | Artificial Intelligence |
| NN | Neural Network |
| DL | Deep Learning |
| ANN | Artificial Neural Network |
| ML | Machine Learning |
| SA | Search Algorithm |
| RNN | Recurrent Neural Network |
| CNN | Convoluted Neural Network |
| FNN | Fuzzy Neural Network |
| DNN | Deep Neural Network |
| PCA | Principal Component Analysis |
| DT | Decision Tree |
| SVM | Support Vector Machine |
| PSO | Particle Swarm Optimization |
| RF | Random Forest |
| KNN | K-Nearest Neighbor |
| SOM | Self-Organizing Map |
| ANFIS | Adaptive-Network-based Fuzzy Inference System |
| GA | Genetic Algorithm |
| GP | Genetic Programming |
| BPNN | Backpropagation Neural Network |
| WNN | Wavelet Neural Network |
| RSM | Response Surface Methodology |
| RBF | Radial Basis Function |
| MLP | Multi-Layer Perceptron |
| SVR | Support Vector Regression |
| GRU | Gated Recurrent Unit |
| LSTM | Long Short-Term Memory |
| IoT | Internet of Things |
| LSSVM | Least Square Support Vector Machine |
| GA-SVR | Genetic Algorithm–Support Vector Regression |
| ELM | Extreme Learning Machine |
| $R^2$ | Coefficient of Determination |
| MSE | Mean Squared Error |
| SSE | Sum of Squared Error |
| RMSE | Root Mean Square Error |
| MAPE | Mean Absolute Percentage Error |
| MAE | Mean Absolute Error |
| WWT | Wastewater Treatment |
| WWTP | Wastewater Treatment Plant |
| MBR | Membrane Bioreactor |
| UASB | Up-flow Anaerobic Sludge Blanket |

| | |
|---|---|
| AGS | Aerobic Granular Sludge |
| TMP | Transmembrane Pressure |
| BOD | Biological Oxygen Demand |
| COD | Chemical Oxygen Demand |
| DO | Dissolved Oxygen |
| TN | Total Nitrogen |
| TP | Total Phosphorus |
| HRT | Hydraulic Retention Time |
| SS | Suspended Solid |
| TSS | Total Suspended Solid |
| CTSS | Color Total Suspended Solid |
| MLSS | Mixed Liquor Suspended Solid |
| MLVSS | Mixed Liquor Volatile Suspended Solid |
| TOC | Total Organic Carbon |
| TIC | Total Inorganic Carbon |
| OLR | Organic Loading Rate |
| TDS | Total Dissolved Solid |
| RT | Reaction Time |
| SRT | Sludge Retention Time |
| EC | Electrical Conductivity |
| EPS | Extracellular Polymer |
| MB | Methylene Blue |

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
