# Peer review of "A Review on Applications of Artificial Intelligence in Wastewater Treatment"

_sustainability, doi:10.3390/su151813557_

Round 1
Reviewer 1 Report
Dear Editors:
Thank you very much for the recommendation to review the manuscript entitled " A review on applications of artificial intelligence in wastewater treatment ". I would like to send you my comments on it.
Revision
In this review article they discuss AI applications in wastewater treatment (WWT). The authors explain commonly used AI models and their applications, advantages and disadvantages, as well as the main findings of particular AI applications related to water quality monitoring, laboratory-scale research to process design.
Suggested changes or recommendations
In line 257, "heavy metals, ..., microbial contamination, drugs and pesticides" is mentioned, however, the examples do not address them, the studies carried out on the application of AI should be studied in depth, with these parameters.
In this review article they discuss AI applications in wastewater treatment (WWT). The authors explain commonly used AI models and their applications, advantages and disadvantages, as well as the main findings of particular AI applications related to water quality monitoring, laboratory-scale research to process design.
It is suggested to review the focus of the study AI not only includes ML, within ML, there are Artificial Neural Networks, deep learning Search, classification and regression Algorithm, so the study rather focuses on ML, as a branch within the AIs.
The authors could also address in more depth the current limitations of implementing WWT with AI, and what is their future vision towards that topic.
Other topics of interest to be discussed could be how to implement AI in image recognition, for example, microbiological water quality tests, microplastics identified by FTIR, SEM, Raman spectroscopy, etc., which are mentioned but not sufficiently exemplified, nor the clear perspective of the authors is addressed
Some minor changes are suggested:
· Figure 1. Classification of AI models for WWT. Revisar la originalidad de las imágenes y los derechos de autor que sean conferidos para su uso.
· Bibliographical references should not be included in the conclusions.
Good!!
Reviewer 2 Report
The manuscript is focused on the applications of artificial intelligence in wastewater treatment. In general, the presented work is interesting. However, before the publication, it should be improced significantly. For this purpose, please, see the comments below:
1. The manuscript is prepared carelessly and it is not in line with the Journal template.
2. The aim of the work should be better described, not just in 1 sentence.
3. The submited manuscript is a review work. However, its novelty should be presented. What is the difference between this work and presviously published papers related to this topic?
4. The work is focused on the wastewaters. Undoubtedly, the work lacks a detailed description of:
- the issues related to wastewaters treatment,
- various wastewaters characteristics.
Such topics should be presented in order to introduce the reader to the analyzed issues.
5. The quality of the Figure 2 should be improved.
6. Section 3.1.1. Laboratory-scale research:
- The Authors wrote about membrane bioreactors, however, there is no precise description of such solution.
- The phenomenon of membrane fouling should be described in more detail. It is a serious problem, which should be discussed in order to introduce the Reader to the presented topic.
7. The literature review should be performed once again. Indeed, it is a review wok and the Authors cited only 108 papers. In order to better described membrane bioreactors and fouling phenomenon as a serious problem limiting the possibilites of using MBRs I recommend the following, recently published papers:
DOI: https://doi.org/10.3390/en15144981
DOI: https://doi.org/10.3390/membranes13020181
DOI: https://doi.org/10.3390/membranes11110887
8. The English should be improved.
The English should be improved.
Reviewer 3 Report
Dear Authors,
Kindly find below the review comments on your manuscript titled: “A review on applications of artificial intelligence in wastewater treatment”
They are:
1. The article is well structured.
2. Line 5 (abstract): The space bar between “performance” and the comma, should be removed.
3. There are English language and coherence issues in the review paper. It requires proof reading and improvement. Text formatting and style needs to be consistent throughout the text.
4. The novelty and practical applicability of this study should be highlighted more in the introduction section. The introduction section could be improved.
5. Authors tried to present the structure of the article at the end of the introduction. However, it is not well structured. Kindly improve on that.
6. The methodology adopted for the review was not highlighted. Thus, it is strongly suggested to create a section describing the methodology adopted, how referenced articles were sourced and from where. The methodology should be more specific and systematized with references to support it.
7. As a review paper, Table of comparisons provided are good. However, it is suggested to add a column on “remarks” to Tables 1, 3, 4, and 5. The column should summarize key findings and recommendations from each reference cited.
8. A section should be provided separately for “Future perspectives”. It is suggested that the future perspective be separated from the conclusion.
9. Line 462: The reference [7] should be removed in the conclusion section. No reference is allowed.
10. Authors should present a section in the manuscript comparing the performance of wastewater treatment plants “with” and “without” the utilization of Artificial Intelligence (AI).
There are English language and coherence issues in the review paper. It requires proof reading and improvement. Text formatting and style needs to be consistent throughout the text
Round 2
Reviewer 1 Report
No comments. Approve as presented.
Reviewer 2 Report
The manuscript has been improved according to my comments, hence, I recommend it for publication.
Reviewer 3 Report
Authors have significantly improved the revised manuscript